# Application of Digital Image Correlation in Structural Health Monitoring of Bridge Infrastructures: A Review

Mohammed Abbas Mousa [1,2], Mustafasanie M. Yussof [1,*], Ufuoma Joseph Udi [1], Fadzli Mohamed Nazri [1], Mohd Khairul Kamarudin [3], Gerard A. R. Parke [4], Lateef N. Assi [5] and Seyed Ali Ghahari [6]

1   Department of Civil Engineering, Engineering Campus, Universiti Sains Malaysia, Nibong Tebal 14300, Malaysia; mohammed.mousa@qu.edu.iq (M.A.M.); udiufuomajoseph@student.usm.my (U.J.U.); cefmn@usm.my (F.M.N.)
2   Department of Roads and Transportation, University of Al-Qadisiyah, Al Diwaniyah 58002, Iraq
3   Faculty of Civil Engineering, Universiti Teknologi MARA, Shah Alam 40450, Malaysia; mkhairul3965@uitm.edu.my
4   Department of Civil and Environmental Engineering, University of Surrey, Surrey GU2 7XH, UK; g.parke@surrey.ac.uk
5   Department of Civil Engineering, Mazaya University College, Nasiriyah 64001, Iraq; lassi@email.sc.edu
6   Department of Civil Engineering, Purdue University, West Lafayette, IN 47907, USA; sghahari@purdue.edu
*   Correspondence: cemustafa@usm.my

**Abstract:** A vision-based approach has been employed in Structural Health Monitoring (SHM) of bridge infrastructure. The approach has many advantages: non-contact, non-destructive, long-distance, high precision, immunity from electromagnetic interference, and multiple-target monitoring. This review aims to summarise the vision- and Digital Image Correlation (DIC)-based SHM methods for bridge infrastructure because of their strategic significance and security concerns. Four different bridge types were studied: concrete, suspension, masonry, and steel bridge. DIC applications in SHM have recently garnered attention in aiding to assess the bridges' structural response mechanisms under loading. Different non-destructive diagnostics methods for SHM in civil infrastructure have been used; however, vision-based techniques like DIC were only developed over the last two decades, intending to facilitate damage detection in bridge systems with prompt and accurate data for efficient and sustainable operation of the bridge structure throughout its service life. Research works reviewed in this article demonstrated the DIC capability to detect damage such as cracks, spalling, and structural parameters such as deformation, strains, vibration, deflection, and rotation. In addition, the reviewed works indicated that the DIC as an efficient and reliable technique could provide sustainable monitoring solutions for different bridge infrastructures.

**Keywords:** bridges; digital image correlation (DIC); vision-based method; structural health monitoring (SHM)

## 1. Introduction

It is common knowledge that civil structures, including bridges, start to deteriorate after being constructed and used. Maintaining efficient and secure structures like bridges for everyday use is essential for the well-being of everyone. Evaluating and ascertaining the state of ageing bridge infrastructure is paramount in verifying structural reliability, ensuring long-term functionality, and deducing when structural components should be repaired or replaced. Tracking and deducing structural reliability and ascertaining the level of deterioration in civil infrastructure is commonly described as Structural Health Monitoring (SHM). Structural health monitoring of civil structures comprises defining, by measured parameters, the area and extent of deterioration in civil infrastructure when this occurs. Structural health monitoring entails observing the identified structure over a given period, extracting susceptible damaged properties to deduce its present state of health, and estimating the long-term condition of the structure [1]. Structural health monitoring

performs a critical function in deterring severe damage, improving structural safety, and minimising maintenance-related downtime and cost.

Structural health monitoring is frequently performed in tandem with yet another directly associated discipline, namely Non-Destructive Evaluation (NDE). These techniques comprise certain assessment procedures used to evaluate the state of a specified structure without compromising the functionality of the structure [2]. Both SHM and NDE techniques enable timely identification and evaluation of infrastructural deterioration, ensuring the system continually meets life safety demands.

Because individual visual examination methods depend on the inspector's judgment, which is labor-intensive and subject to variations, it makes computerised surveillance devices extremely desirable. Contact-based devices are often utilised to track the number of infrastructure components. Previously, the standard trend was to collect and disseminate information from a host of devices embedded in the targeted system [3]. Such approaches, including both static and dynamic assessment, have grown markedly. Contact-based devices, which include transducers [4–6], strain gauges [7,8], inclinometers [9,10], accelerators [11] and extensometers [12], are often utilised for SHM applications. Other alternatives are fibre optic sensors, which have been demonstrated as credible options to traditional sensors because of their versatility, scalability, and Electro-Magnetic Interference (EMI) [13–16]. However, such sensors can be quite challenging to execute, requiring cabling and power; they are also expensive and are typically not flexible for several infrastructural components when they are installed. Several researchers have suggested using a Wireless Sensor Network (WSN) to address these issues [17,18].

Nonetheless, it seems that these sensors are usually not sufficiently robust to be mounted or installed on the targeted structural component and conduct assessments throughout the service life of the infrastructure, which can be years or decades depending upon its construction and when defects are much more likely to be present. Besides, some of these sensors may supply data only at discrete points [19]. Subsequent technological advancements further produced ample NDE techniques for evaluating various engineering applications. Radiography [20], radioactive computerised tomography [21], radar [22], infrared thermography [23], ultrasonic arrays and acoustic imaging systems [24], and Acoustic Emission (AE) [25] have already been applied for both SHM and NDE, each having its strengths and limitations. The recent development of camera capabilities, vision sensors, and image analysis algorithms enabled the latest innovations of non-contact tracking techniques. Photogrammetry and vision-based methods utilise image tracking with optical devices to detect the precise location of points and features and utilise this information to analyse their movement across various periods or phases. Vision-based methods have proven to be an efficient and effective strategy for non-contact analysis and structural deformation extractions, full-field displacement, geometry profiles, and strain in civil applications [26–29].

## 2. Digital Image Correlation

Digital Image Correlation (DIC) is a vision-based technique that utilises image detection and mapping technology to precisely measure variations of images in 2D and 3D. DIC is commonly used in many fields of science and engineering. It is frequently employed to analyse full-field displacement and strain, crack propagation, identifying material deflections, damage emergence, and progression in structural systems. DIC can be employed for routine or long-term surveillance with photographs of the targeted system taken at varying times. Images from the different periods can be examined optically with software, and the deformations can be measured from the data [30]. Many vision-based algorithms have been suggested, which include edge detector [31], thresholding [32], segmentation [33], and filter-based algorithms [34]. These techniques have been utilized in assessing damage in wood samples [35–37], bridge and concrete surfaces [38–40], pavement [34,41,42], and steel [43–45]. Other applications of DIC include biomechanical applications [46], biological tissues and biomaterials [47–49], aerospace [50,51], computer vision [52], medical appli-

cations [53], industrial and automotive applications [54]. In contrast to other traditional Non-Destructive Techniques (NDT) measuring methods and specific visual methods like radioactive computerised tomography, which are otherwise costlier and quite challenging to employ beyond the laboratory environment because they entail detailed setup and low vibrational surroundings, DIC is economically feasible and easier to grasp. DIC is more quantifiable and precise than the traditional measuring techniques [55]. In addition, DIC employs traditional electronic imagery, which could be utilised with civil construction methods to produce sufficiently precise assessments of component systems in conventional outdoor surroundings. Furthermore, no specialised illumination is needed, and in certain instances, DIC can function without any peculiar surface prepping due to the targeted system existing layer having adequate photographic texture.

The Digital Image Correlation (DIC) basic concept compares the captured images at varying levels of deformation of a targeted system and analyses them using correlation-based matching algorithms. This technique can be used to calculate material deformation by monitoring a neighbourhood of pixels grey-level and creating full-field deformations in 2D and 3D strain charts and vector fields. Photographs are obtainable across a broad spectrum of sources ranging from high-speed video recorders and portable commercial cameras to the traditional Charge-Coupled Device (CCD). Any structural differences can be accurately contrasted with the captured photographs, and thus, unforeseen changes that could induce irregularities would be effortlessly detected. An illustration of the application of DIC is the calculation of crack width and crack detection. If a targeted component were to undertake load crack checks like in the reinforced concrete beam shown in Figure 1, the position of the major vertical crack would be quickly identified. Nonetheless, there could be several minor cracks that are not easily visible to the naked eye. As illustrated in Figure 2, utilising a marker pen enables the identification of such cracks, although a time-consuming process and in a disorderly manner.

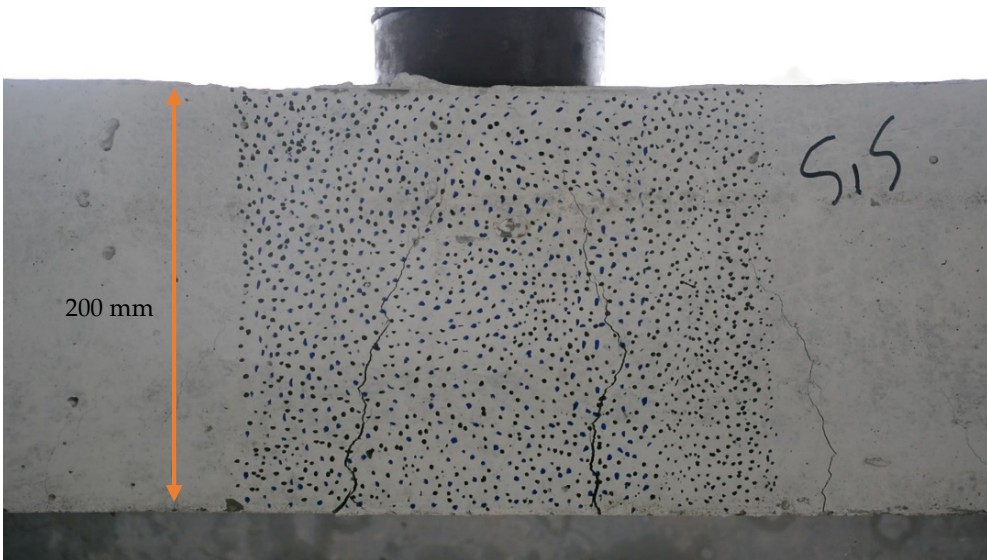

**Figure 1.** A reinforced concrete beam after load testing revealed two major vertical cracks within the speckled region, and no noticeable minor cracks (images by Mohammed Mousa).

By employing DIC, as illustrated in Figure 3, the total actual size and number of cracks can be identified, inside the speckled zone, from a single captured photograph before cracking emerged and another captured afterwards. This technique provides advantages in detecting cracks, which may develop in a non-contact manner. Furthermore, it offers valuable data about where crack propagation detectors can be attached and proffer precise calculations of cracks, including instances where crack edges are poorly defined [56].

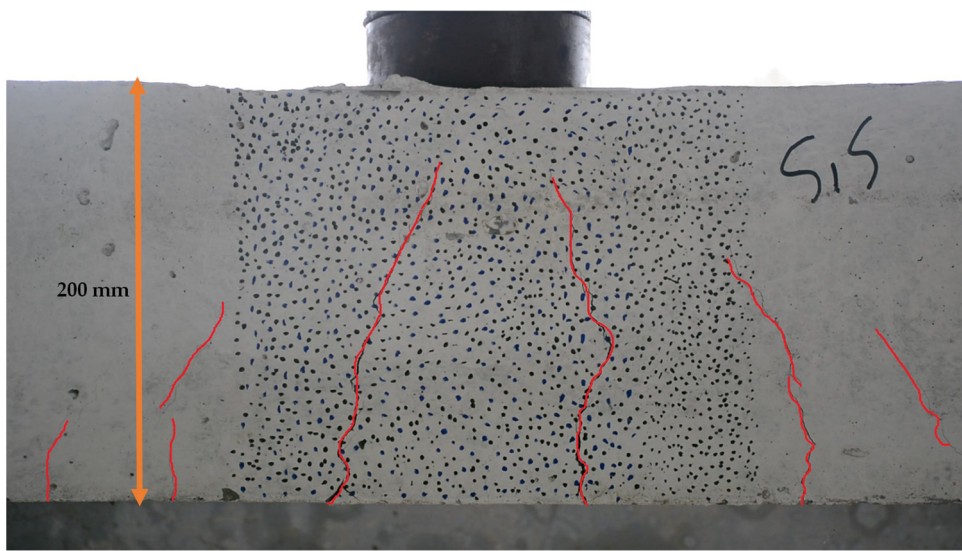

**Figure 2.** Marker pen identifying minor cracks which are not detectable by the human eye (images by Mohammed Mousa).

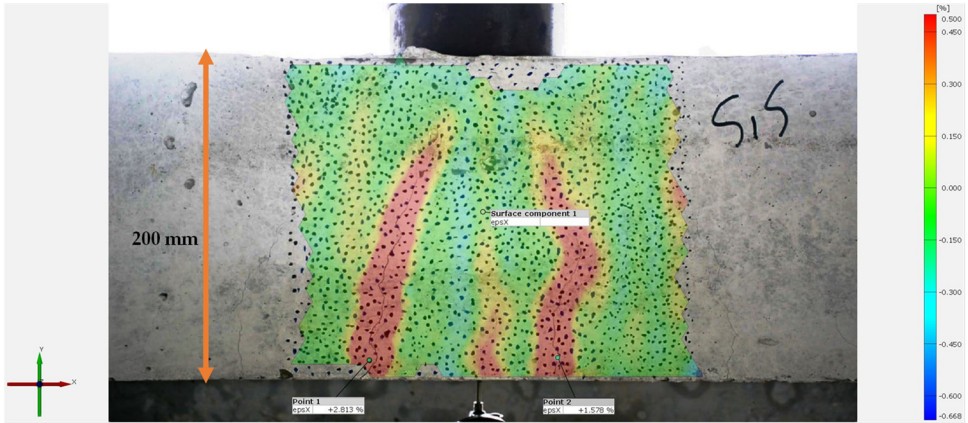

**Figure 3.** Digital Image Correlation (DIC) deformation map reveals barely visible cracks by the naked eye; notice the third crack between the two major cracks not identified by the marker method (images by Mohammed Mousa).

The practice of employing cross-correlation in quantifying changes in data algorithms has long been discovered and has been applied to electronic imagery since the 1970s [57,58]. The earliest studies, the foundation, and framework in DIC methods in mechanics were inspired by researchers in the 1980s [59–61]. Subset-based DIC techniques such as 2D and 3D concepts were developed during this period and, since then, have not experienced drastic alterations until recently. Due to the advancement of computer computational capabilities, a new global finite element-based approach for the DIC correlation algorithm has been developed [62–64]. New DIC trends that are hybrid-based, as well as machine learning and artificial intelligence-centred approaches, have emerged [65–67].

### 2.1. Two-Dimensional Digital Image Correlation

Using 2D-DIC techniques that require a sole digital device remains restricted to calculating the deformation of planar or flat systems. By contrasting photographs of the targeted system captured at multiple periods or at varying levels of deformation via the subset-based or finite element method, the material displacements in strain charts and vector fields could be explicitly extracted [68]. Two-dimensional DIC (see Figure 4)

techniques do not apply in situations where the targeted system has a curved surface or, in instances whereupon loading, 3D deformation emerges [69].

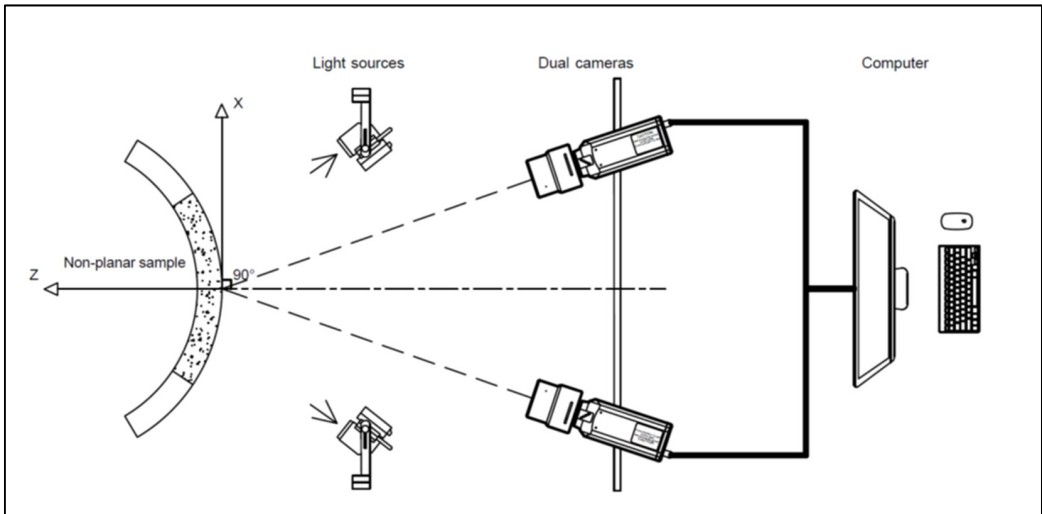

**Figure 4.** 2D-DIC planar surface deformation measurement, side view.

Like any other technique, the DIC techniques have limitations related to either machines or humans. These limitations can be summarised by (1) optical distortion for various reasons, (2) distance change between the lens and the observed surface, which can be caused by horizontal or vertical camera tilting, specimen tilting or both, (3) out of plane local bulging of the specimen, and (4) as it is a vision-based technique, DIC is only able to measure deformations and strains at the surface of the specimen.

There are several errors produced by the 2D-DIC technique, the most common one being the cosine error produced by either camera or specimen tilting, leading to the measurement of deformations that are higher or less than the actual deformation. This error can be treated by calculating the tilting angle, then emerging into a particular equation to calculate the error percentage. Other errors can occur, such as gross displacement, produced by either camera or specimen gross movement while testing. This error can be treated by adding or subtracting the produced deformation by the movement, which can easily be detected by observing the deformation or strain curves. However, such a process sometimes becomes tedious when dealing with complicated or large structures; therefore, a 3D-DIC technique can overcome such difficulties.

### 2.2. Three-Dimensional Digital Image Correlation

A 3D-DIC approach was later devised to address this obvious limitation of the 2D-DIC technique by employing two synchronised digital cameras, as shown in Figure 5 [63]. A 3D-DIC would then precisely quantify 3D deformation in a vector field of both curved and planer surfaces on the premise of the binocular stereo-vision concept, enabling a versatile and effective technique in practical applications. To execute the 3D-DIC technique, the configuration of the binoculars will be performed first to find out both constitutive and supervenient boundary conditions of the synchronised digital cameras. Three-dimensional deformations at measured levels could then be reproduced based on configured metrics discrepancy charts measured with DIC algorithms [70].

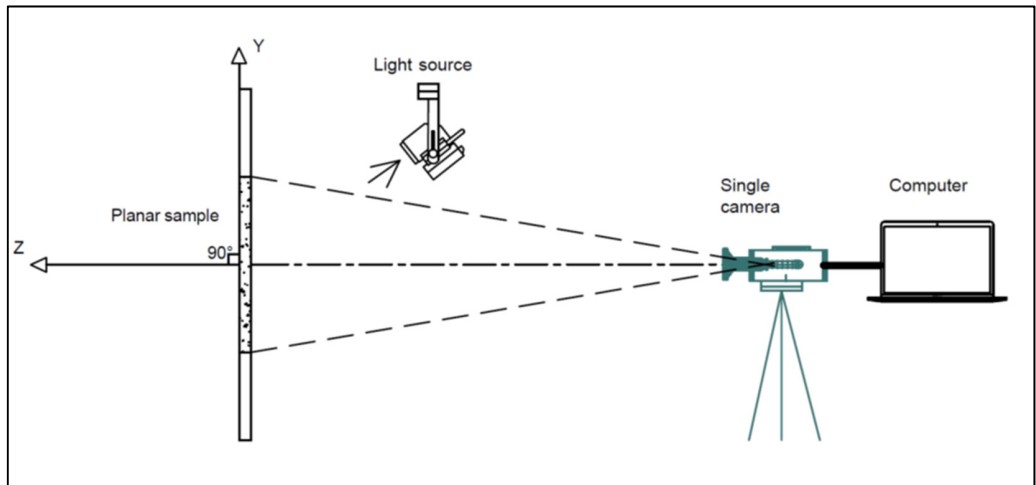

**Figure 5.** 3D-DIC curved surface deformation measurement, top view.

Following the trail developed by early scholars and accelerating the technological prowess of advanced virtual devices, recent research endeavours have been devoted to implementing increasingly reliable, more sustainable, and rigorous DIC methods. Besides the typical imagery devices, diverse and innovative imagery and video devices, as well as equipment such as Scanning Electron Microscopes (SEM) [71], high-speed and ultra-high-speed cameras [72], infrared cameras [73], x-ray cameras [74–77], advanced video extensometers [78], high accuracy video extensometers [79], drones and Unmanned Aerial Vehicles (UAVs) [28], have all been newly proposed to measure real-time deformations and high-precision strain computation at different temporal and geometrical scales for the categorisation and resolution of irregularities induced by previously employed conventional equipment.

Furthermore, significant progress has been achieved in several areas of 2D- and 3D-DIC, such as the optimisation of speckle patterns that serve as quality evaluation criteria and transmitter of deformation data on tested specimens [80–83]. Other advancements and insights were dedicated to the correlation algorithms, including Finite Element Method (FEM)-based. The new correlation algorithms precisely compute discontinuous displacements detected in objects bearing cracks and ascertain a similarity or disparity index between a referenced test object and its deformed counterpart [84,85]. Moreover, numerous attempts have been made to further improve the reliability of the traditional subset methods in broadening DIC preview to increasingly demanding surroundings such as high-temperature or outdoor surroundings [86] and greater comprehension of fracture properties, as well as a clearer understanding of fracture dynamics at various levels of deformation [87]. The method of synchronising two ultra-high-speed cameras in addressing growing requirements of deformation computation in 3D full-field high-speed has also been devised and extensively studied [88]. Supported by a specially built light-dividing appliance, the one-camera 3D-DIC method has been developed [89]. The suggested technique negates the demand for two accurate synchronised cameras. Further, the technique significantly minimises hardware expenditure. In addition, the multi-camera 3D-DIC method capable of capturing the test object's surface photographs from multiple angles by employing multiple cameras has garnered much interest in recent years [90]. Moreover, there has been a well-established and widely acceptable standardised technique in camera calibration for achieving top-notch stereo-vision and, thus, accurate 3D computation [91].

*2.3. Digital Volume Correlation Method*

A 3D expansion of the 2D-DIC technique, the Digital Volume Correlation (DVC), has also been a recent development of DIC. The DVC technique (see Figure 6) emerged as an effective medium in measuring intrinsic deformations of opaque components like

wood [92], bony and skeletal specimens [93] and rock [75] as a result of comparing volumetric photographs obtained from 3D photographic devices. The DVC technique was firstly introduced by Bay et al. in 1999 as a tool to find a continuum-level deformation in trabecular materials such as bones [74]. The applications of DVC are mainly used for a high level of accuracy measurements, such as micro-strains produced in biological specimens such as bones. The DVC also has some limitations, such as the limitation of the continuum approach to measuring local strains in foam materials at high levels of strains [74]. This error can be remedied by adding rotation and strain parameters in order to obtain more accurate measurements of the local effects.

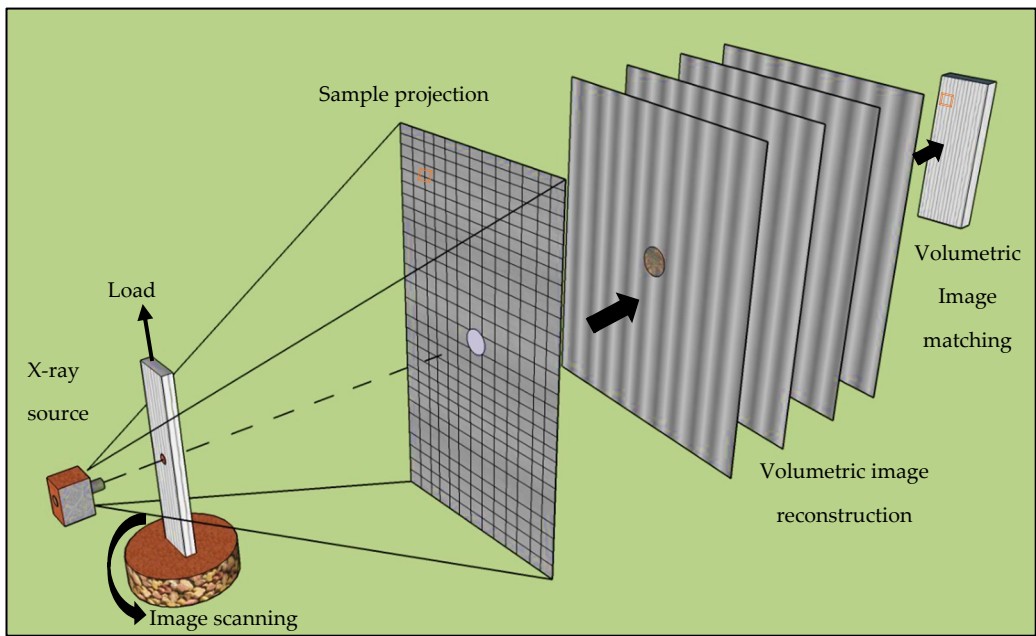

**Figure 6.** Digital Volume Correlation (DVC) dog-bone 360° surface deformation measurement, isometric view.

## 3. Structural Health Monitoring Using DIC in Bridge Structures

An essential approach to long-term civil infrastructural cost-effective management is to employ minimal-cost monitoring methods that could be easily implemented and sufficiently precise. The Digital Image Correlation (DIC) satisfies these requirements. As a result of their immense sizes, DIC provides an ideal solution to the traditional sensors for civil, structural monitoring because they enable assessments to be carried out without any interference with the system functionalities and could access locations that are challenging to reach without the use of special machinery, like scaffolds and trucks. During the last two decades, many researchers realised the efficiency and versatility of the vision-based and DIC techniques for various SHM applications. Because of their strategic significance and security concerns, bridges are the mostly surveyed civil engineering systems. Following is a state-of-the-art review of the latest SHM application using the DIC technique for multiple bridge infrastructures. A summary of the recent DIC applications in SHM of bridge structures is presented in Table 1.

### 3.1. Concrete Bridges

Concrete bridges are divided into several types: Arch, Reinforced Slab, Beam and Slab, Box Girder, Integral, and Cable-Stayed Bridges [94]. One of the earliest works conducted by Lee and Shinozuka in 2006 was applied to a concrete bridge with steel box girders, which will be later described in the steel bridge section [95]. In a study, a 2D-DIC technique was employed by Yoneyama and Ueda in 2012 in tracking the variance of a bridge girder's vertical deflection by proposing a method for correction of camera movement influence [96].

In the experimental study, a heavyweight was mounted on a bridge. A single electronic camera captured photographs before and after the deformation of the girder surface, as shown in Figure 7. The findings revealed that the DIC-recorded deflections agreed with those observed with the Linear Variable Differential Transformer (LVDT), suggesting the proposed technique could eliminate the influence of camera motion [96].

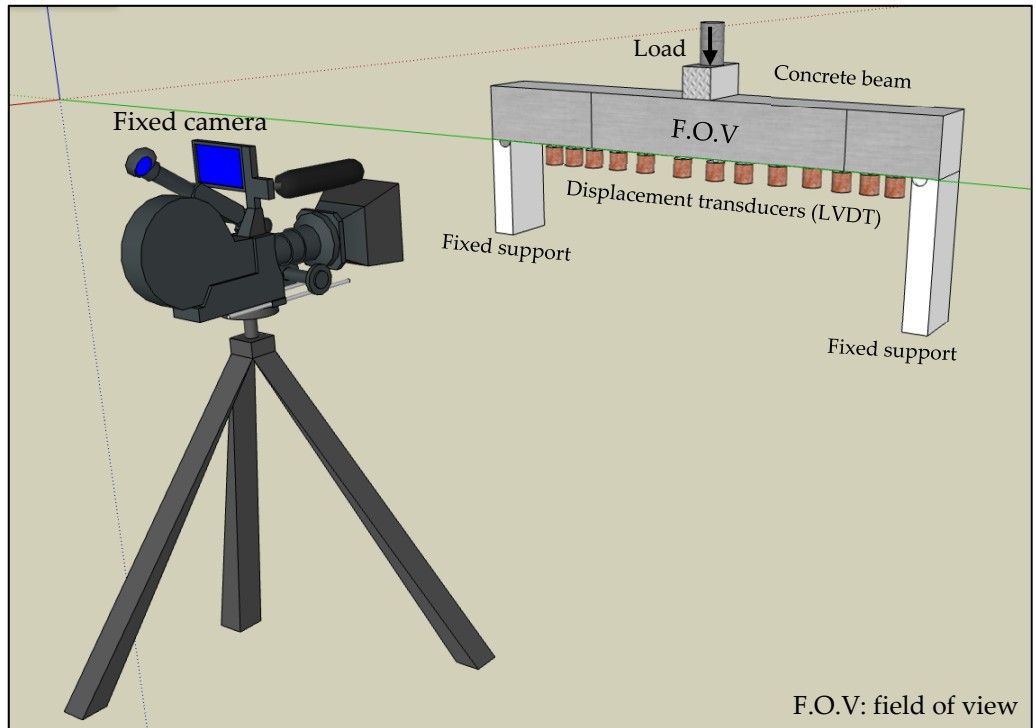

**Figure 7.** Concrete bridge girder experimental setup using 2D-DIC [96].

Using the 2D-DIC technique, Feng et al., in 2015, extracted displacements in real-time from video images by devising an advanced prototype pairing algorithm called up-sampled cross-correlation using Fourier transform [97]. The results showed that by merely altering the up-sampling parameter, improved imagery resolution of less than 1 mm could be accomplished effortlessly. The field test carried out on a railway and pedestrian bridge showed that the displacements measured by the single camera agreed with measured displacements obtained from the high-performance laser sensors. The vision-based technique gives significant advantages such as efficiency in extracting structural displacements from a single measurement at any point, besides its ease of deployment and low cost [97].

In a similar study, Pan et al. in 2016, proposed an off-axis 2D-DIC technique by employing an inverse Gauss-Newton compositional algorithm for real-time vertical displacement measurement of a rail bridge girder with the aid of an advanced video deflectometer [55]. The experimental outcomes indicated that the proposed method could detect planar displacement due to the pin-hole digital camera model used for measuring the in-field bridge deflection and provides attractive benefits such as targetless and truly remote sensing and visualisation of measurements in real-time [55].

A three-dimensional DIC was also utilised by Nonis et al. in 2013 to examine a concrete bridge consistently to detect invisible cracks, assess spalling, and evaluate the deformation of the bridge [98]. The experimental results indicated that the photographic analysis aligned with that conducted while utilising fibre optic strain gages across three and four-point bending tests performed on a concrete beam [98]. The same bridge was surveyed by Reagan et al. in 2017, using 3D-DIC for nearly a year [30]. The authors suggested a new technique incorporating the utilisation of 3D-DIC and an Unmanned Aerial Vehicle (UAV), which will conduct health monitoring of bridges utilising the non-contact, optical-based assessment method. The authors revealed the effectiveness of this technique in identifying

structural alterations and tracking the dynamic activities of expansion joints and hairline cracks over time [30].

Winkler and Hendy, in 2017, utilised DIC to monitor and assess a single-span simply supported Warton bridge constructed in 1937 [99]. The DIC was implemented on the bridge as a part of the fatigue assessment strategy by running three-car trains over the bridge deck and measuring deflection. The DIC is comprised of a single camera mounted on a tripod at 30 m from the bridge, with a sampling frequency of 30 Hz and a resolution of 1/50th of a pixel. Using the DIC showed a cost-effective and easy-to-implement approach for strain measurement and structural performance monitoring of the Warton Road concrete bridge [99].

Recently, a new approach has emerged based on an off-axis DIC video deflectometer as a robust yet straightforward contactless vision-based technique, proposed by Tian et al. in 2021 [100]. The new approach was intended to measure a full-field deflection over many points, utilising the "straight-line fitting scheme". Furthermore, the suggested approach uses reliability-guided processing and a quick initial parameter estimate procedure for real-time and accurate image-matching analysis. The proposed technique was applied to both an indoor cantilever and a high-speed railway concrete bridge to demonstrate the technique's effectiveness and practicality for SHM applications. It was found that the accuracy of the proposed approach mainly depends on multiple factors such as the field of view, camera resolution and sampling, measurement distance, concrete surface texture, and environmental conditions [100].

*3.2. Suspension Bridges*

Using DIC, monitoring hanger cables is another common application for civil infrastructures, especially cable-stayed bridges. Ji et al., in 2008, were among the earliest researchers who envisioned the use of combined 2D- and 3D-DIC techniques to observe vibrations of cables holding a pedestrian bridge utilising optical sensors [101]. Utilising a single commercially accessible camera device with no camera calibration prior to testing is required; the suggested approach studied a series of photographs taken from a vibrating wire and computed the intensity of optical variance of a targeted area of reference to determine the wire displacement. The experimental findings indicate that the approach could precisely quantify cable vibration and pipe motion. Furthermore, this optical flow approach allows for simple implementation by circumventing the necessity of appending a target on the cable [101].

Kim and Kim employed Digital Image Correlation (DIC) in 2013 to create a photo correction algorithm to examine photographs obtained with an electronic photo device to measure the dynamic characteristics of the bridge's hanger cables stress by calculating the dynamic behaviour of the Gwangans' bridge hanger cables [102]. In both stress and modal frequencies, the laboratory results gathered while utilising the DIC technique and those obtained while utilising the accelerometer-based technique were distinguished by a discrepancy of ±0.5% [102].

A novel "optico-acoustic" approach was presented by Vanniamparambil et al. in 2013 to detect breaking in seven-wire steel cable strands [103]. The newly implemented method combined three Non-Destructive Techniques (NDT), consisting of the Acoustic Emission (AE), Guided Ultrasonic Waves (GUW), and the emerging vision-based Digital Image Correlation (DIC). The integration of these NDT techniques simultaneously was intended to enable cross-correlation and validation of the damage detection, initiation, and progression of the steel wire. The applied approach showed a remarkable reduction of the uncertainties associated with each technique due to the concurrent use of these techniques [103]. The execution of these techniques for wire damage detection shows the effectiveness of using DIC for SHM systems, especially when combined with other heterogeneous sensing modes like the "optico-acoustic" approach.

Du et al., in 2020, investigated the dynamic response of stay cable Guanhe bridge by using two measurement techniques, Digital Image Correlation (DIC) and Digital Im-

age Processing (DIP), together with the use of the traditional accelerometer [104]. Two laboratory experiments, including single and multipoint cable force measurement, were performed and compared with the accelerometer and the numerical simulation to assess the validity of digital image methods. The findings of the experiments showed only a 5% relative deviation between the DIC values and the accelerometer measurements. It has been concluded that DIC use for cable-stayed bridges provides more sustainable and efficient approaches than traditional contact-based techniques [104].

A new tool using an Unmanned Aerial Vehicle (UAV) was recently proposed to monitor Nansha suspension bridges cables by Tian et al. in 2021 [105]. The technique was employed to capture the cables' vibration images, extract the dynamic characteristics and analyse them using image analysis. The proposed image analysis technique was established based on two aspects, namely (1) the Line Segment Detector (LSD) and (2) frequency difference. The line segment was applied to detect the edges of the cable from the images to extract the dynamic displacement of the cable by proposing a line match algorithm. To circumvent the difficulties of extracting the fundamental frequency from the UAV-obtained video, the frequency difference of neighbouring higher modal frequencies detected from the relative cable displacements was used for force calculation in the cable. The new suggested LSD technique provides a robust approach for the dynamic displacement of suspension bridge cables compared to the traditional image correlation-based algorithms. The proposed system calculated results agree well with those obtained by the accelerometer and fixed camera [105].

### 3.3. Masonry Arch Bridges

Masonry arch bridges comprise around 40% of the bridge infrastructure stock in Europe alone, while most of them have exceeded 100 years of service [106]. Most old masonry bridges assessment and maintenance conditions rely heavily on visual inspection to identify the deterioration and structural behaviour changing during their working life [106]. In addition, the field inspection of masonry bridge structures using the DIC is very rare [107]. Koltsida et al., in 2013, used 2D-DIC at the field to monitor a four-span railway masonry arch bridge under normal traffic conditions [106]. The study was implemented to determine the DIC efficacy for bridge monitoring and assess the bridge structural and service conditions. The DIC setup comprised a single camera placed at a 10 m distance from the observed service. The first and the second span were monitored with the camera when a 45 tonnes per bogie train passed over the bridge. The horizontal and the vertical displacement versus the train locations were calculated, in pixel units, using the MatchID-2D DIC software [108]. The DIC technique proved to be suitable for crack and displacement measurements. In addition, the behaviour changes under loading were easily measured on small-scale specimens, and the results were beneficial for identifying the crack location [106].

A case study of masonry viaduct bridge multipoint dynamic displacement measurements in Leeds, UK, was performed by Acikgoz et al. in 2018, using 2D-DIC [109]. The researcher addressed the limitations related to the DIC technique by surveying and identifying the errors that affect the field DIC measurements. The study also discussed the minimisation and mitigation techniques during the setup (acquisition) and after (post-processing). Among those addressed errors related to the setup (video acquisition) are the camera movement, lens distortion, light exposure, and environmental conditions. On the other hand, errors related to the structural behaviour were identified by the out-of-plane rigid body movement and rotation. Lastly, the errors related to the video processing (post-processing) depend on the target surface quality (pattern quality), target size, and distance from the camera. An arch span of 7.7 m, 2.1 m height, and 8 m width, having a half-meter thick barrel, was investigated in this study. The bridge is designed to carry two tracks of 25-ton axels trains at a 55 km/h speed. The field setup comprises two commercial cameras with a resolution of 2 MP and a sensor size of $12 \times 6$ mm at a 50 Hz frame rate. The video processing of the DIC was implemented using the Imetrum software. It has been concluded

that quantifying errors produced by the out-of-plane movement using a simple geometry technique with the aid of the pin-hole imaging model will provide helpful information about the out-of-plane effects on the in-plane DIC measurements [109,110].

Recently, Dhanasekar et al., in 2019, investigated the serviceability (deflection and strain) of two old masonry arch bridges in Australia using DIC [107]. Three key sections were selected on the arch bridge to be monitored, namely, quarter, support, and crown. The bridge was monitored under the axle load of two different passenger and freight trains. Three independent cameras of 2.35 MP situated at a 4 m distance from the surface were used to capture digital images of each section for two days. An Istra4D commercial software was used for the DIC analysis and acquisition at 0.02-s intervals [111]. The acquired parameters using the DIC were the deflection, radial, and tangential (axial) strains. In addition, a numerical finite element model was also established to validate the field DIC measurements. Results from the DIC technique show its suitability for field testing, despite the bridges' low deflection and strains values. In addition, a maximum difference of 14.5% was identified between the numerical and corresponding field measurements for the case of tangential strain at the compressive region of the quarter section [107].

### 3.4. Steel Bridges

Steel bridges are broadly used all around the world in multiple structural forms. Steel bridges are preferred over other concrete or masonry bridges due to their high material strength, ductility, and ease of fabrication and construction. One of the earliest works conducted by Lee and Shinozuka in 2006 was applied to a concrete bridge with steel box girders [95]. The vision-based system was proposed to assess the dynamic displacement of the bridge's girders in real-time. The technique was implemented by employing a telescopic appliance with a video camera, a method for image processing requiring a targeted identification algorithm. The experimental results indicated that the recommended model could quantify displacements with accuracy, which was less than a three percent deviation in peak value compared to when utilising the Linear Variable Differential Transformer (LVDT). The key distinction between this study and prior research is the utilisation of consumer-grade cameras with no specifically designed optical modules and the omission of an offline specialised transceiver extraction algorithm [95].

Peddle et al., in 2011, employed DIC to calculate the deflection of a three-span steel girder bridge in Vernon, Massachusetts, US [112]. The bridge length is 47 m built with six steel girders designed to carry a 20 cm concrete deck. The camera setup used for DIC was configured at a sampling rate of 1 Hz, used in conjunction with LVDT having a sampling rate of 50 Hz. The load was applied by three axles truck to apply a force of 320 kN to stimulate the bridge response under traffic load. Results from the load test show an excellent agreement between the DIC and the LVDT.

In addition, the study also deployed the DIC technique for deflection measurement of the concrete bridge supported by steel beams while retrofitting in Gilford, New Hampshire, US. The retrofitting process included removing all the beams and installing FRP strips on the lower face of the bridge. Speckled steel L-shaped tabs were glued to the lower concrete surface to allow the camera to measure deflection at any point on the bridge. A 120 kN empty truck was used to apply the load on the bridge. It was realised that the truck's weight was not sufficient to generate a noticeable amount of deflection, which was only 0.06 mm. The results also showed that the DIC data at small displacement did not agree well with the LVDT data. The variation of the deflection values between the DIC and LVDT could be attributed to using a small focal length lens of 35 mm during the acquisition and can be mitigated with a larger lens. Nevertheless, DIC still offers more convenient and fast installation techniques for SHM [112].

Winkler and Hendy, in 2017, discussed the implementation of DIC for field monitoring of fatigue-sensitive steelworks details of the London's Dockland railway bridge [113]. The bridge consisted of a two-span of 11.7 and 12.7 m. Each span has seventeen longitudinal steel girders. The DIC setup comprised a camera with a sampling rate between 20–40 Hz,

with a resolution of 2% of a pixel. The DIC camera was used to observe a critical section (detail connection) and placed at 0.3 m from the surface. Results from the DIC revealed that high strains were concentrated at the notch detail corner, suggesting that it must be further examined for fatigue cracking. Finally, the field observation demonstrates that using DIC for SHM will facilitate the installation, avoid the disruption of stakeholders, and make no change to the structure's appearance [113].

Recently, Wang et al., in 2019, have investigated the feasibility of pressure-activated tape as a new patterning scheme for coated and bare steel structures [114]. This study utilised a commercial 3M$^{TM}$ pressure-activated adhesive tape with an electronic designed, printed pattern on two steel bridges in Delaware and Indiana, US. The tape has a customised length, 1.5 m width, and has an eight-year durability warranty. The tape application to the structure comprised three steps: surface cleaning, loose coating removing, and tape adherence with bare hands. Besides, the DIC setup was used to monitor the upper chord repairing of the Delaware River bridge. Next, the experimental works include measuring the longitudinal and transverse elastic strains for coated and bare steel specimens.

Moreover, the DIC was utilised with strain gauges, extensometers, and DIC with conventional paint patterning. Results from DIC measurements report lower strain values for the tape specimens compared to the conventional spray pattern. The low strain values could be attributed to the fact that the tape is composed of multiple layers made of different material properties as DIC measures the surface strains; this will measure the strain of the outer layer of the tape and not the real steel strain. Nevertheless, a strain match was observed after a specified length, which is later considered as the "development length" of the adhesive tape. After the development length had been quantified, the measured strain of the steel was correctly predicted as 0.11%. The study recommended checking the conditions of the coating before the use of the tape to avoid any bonding complications while measuring [114].

**Table 1.** Summary of the recent DIC applications in bridge structures' Structural Health Monitoring (SHM).

| Bridge Type. | Description | Location (Year) | Authors and Reference |
|---|---|---|---|
| Concrete | Concrete bridge with a steel box girder | South Korea (2006) | Lee and Shinozuka, 2006 [95] |
| | Concrete girder bridge | Japan (2012) | Yoneyama and Ueda, 2012 [96] |
| | Three full-scale concrete bridges | United States (2013, 2017) | Nonis et al., 2013 [98] Reagan et al., 2017 [30] |
| | Railway and pedestrian concrete bridge | United States (2015) | Feng et al., 2015 [97] |
| | Rail bridge girder | China (2016) | Pan et al., 2016 [55] |
| | Concrete bridge deck | United Kingdom (2017) | Winkler and Hendy, 2017 [99] |
| | A high-speed railway concrete bridge | China (2021) | Tian et al., 2021 [100] |
| Suspension | Cable-stayed pedestrian bridge | Hong Kong (2008) | Ji et al., 2008 [101] |
| | Suspension bridge hanger cables | South Korea (2013) | Kim and Kim, 2013 [102] |
| | Seven-wires steel cable strands | United States (2013) | Vanniamparambil et al., 2013 [103] |
| | Cable-stayed bridge model | China (2020) | Du et al., 2020 [104] |
| | A long-span suspension bridge | China (2021) | Tian et al., 2021 [105] |
| Masonry | Four-span old masonry railway arch bridge. | United Kingdom (2013) | Koltsida et al., 2013 [106] |
| | Masonry viaduct bridge | United Kingdom (2018) | Acikgoz et al., 2018 [109] |
| | Two old masonry railway arch bridges | Australia (2019) | Dhanasekar et al., 2019 [107] |

**Table 1.** *Cont.*

| Bridge Type. | Description | Location (Year) | Authors and Reference |
|---|---|---|---|
| Steel | Concrete bridge with a steel box girder | South Korea (2006) | Lee and Shinozuka, 2006 [95] |
| | Three-span steel girder bridge and concrete bridge supported by steel beams | United States (2011) | Peddle et al., 2011 [112] |
| | Fatigue-sensitive steelworks details of a railway bridge | United Kingdom (2017) | Winkler and Hendy, 2017 [113] |
| | Two steel bridges (coated and bare steel structures) | United States (2019) | Wang et al., 2019 [114] |

## 4. Conclusions

This paper focuses on developing and applying vision-based methods, mainly 2D and 3D-Digital Image Correlation (DIC), for full-field deformation measurement in the Structural Health Monitoring (SHM) of bridge structures. There were four bridge types: concrete, suspension, masonry, and steel bridges, respectively.

The current review demonstrates the DIC potential for SHM applications at multiple bridges at various spans and building materials under different testing and environmental conditions. The bridge sections monitored with the DIC ranged from a few centimetres to multiple meters. Consequently, the versatility of the DIC technique stems from its ability to pair with other modern techniques and tools such as drones, smartphones, and robots. In addition, the non-contact and full-field features of the DIC provided alternative solutions for various situations where traditional monitoring devices cannot be applied. The field applications proved the DIC reliability by comparing the calculated displacements and strains with the conventional point-wise metrological tools, such as the Linear Variable Differential Transformer (LVDT), strain gauges, and numerical simulation. Results variation between the DIC and other techniques were observed and associated with their causing sources. Among the primary sources of data variation are the illuminating light flickering effect, camera stabilisation, and speckle pattern quality. Further, the influence of the DIC fundamental parameters such as camera resolution, camera distance, and acquisition rate on the quality of the measured data were described. Previous works showed that the image quality, sharpness (no-vibration), and speckle pattern contrast play an essential role in the quality of the captured data with the DIC.

Furthermore, the tracing of advances made over the last three decades in DIC, such as the improvement of digital cameras, smartphones, and computational algorithms, contributed to making the DIC a robust and well-established technique. It has been widely regarded as a transformative discovery for full-field deformation computation in static testing and dynamic investigations. Finally, due to the rapid advancement of optical instrumentation, such as high-definition video recorders and unmanned autonomous devices for capturing images and matching algorithms (including FEM software) for analysing the captured images in full-field 2D and 3D, the DIC technique will arguably remain the most effective and versatile tool for deformation measurement in the coming years.

**Author Contributions:** Conceptualisation, M.M.Y. and U.J.U.; methodology, M.M.Y., U.J.U., M.A.M., F.M.N. and M.K.K.; validation, F.M.N., M.M.Y., L.N.A. and M.A.M.; formal analysis, M.M.Y.; investigation, M.K.K. and M.A.M.; resources, M.M.Y.; data curation, L.N.A.; writing—original draft preparation, M.M.Y.; writing—review and editing, M.A.M.; visualisation, M.A.M., F.M.N. and M.K.K.; supervision, G.A.R.P.; project administration, M.M.Y.; funding acquisition, M.M.Y., L.N.A. and S.A.G. All authors have read and agreed to the published version of the manuscript.

**Funding:** This research is funded by the Ministry of Higher Education Malaysia under the Fundamental Research Grant Scheme, grant number 203/PAWAM/6071408.

**Institutional Review Board Statement:** Not applicable.

**Informed Consent Statement:** Not applicable.

**Data Availability Statement:** Not applicable.

**Acknowledgments:** The authors express their acknowledgements to the Ministry of Higher Education Malaysia for their Fundamental Research Grant Scheme.

**Conflicts of Interest:** The authors declare no conflict of interest.

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
