# Peer review of "Application of Digital Image Correlation in Structural Health Monitoring of Bridge Infrastructures: A Review"

_infrastructures, doi:10.3390/infrastructures6120176_

Round 1

Reviewer 1 Report

Journal name: Infrastructures

Manuscript number: infrastructures-1454708-peer-review-v1

Full Title: “Application of Digital Image Correlation in Structural Health Monitoring of Bridge Infrastructures: A review”

Authors: Mustafasanie M. Yussof, Ufuoma Joseph Udi, Mohammed Abbas Mousa, Fadzli Mohamed Nazri, Mohd Khairul Kamarudin, Gerard A.R. Parke, Lateef Assi, and Seyed Ali Ghahari

This work proposes a literature review about the Digital Image Correlation in Structural Health Monitoring of bridge infrastructures.

General comments: the paper is well formatted; figures are clear (even if not all in hi-res); the topic fits well the purposes of the Journal.

The collection proposed by the Authors is of interest from an engineering standpoint and, as far as the knowledge of the Reviewer, the results are new and here published for the first time. Nevertheless, the following questions need to be carefully addressed by the Authors prior to publication.

1) The bibliography must be improved with recent works: out of 106 total articles, there are only four papers from 2020 and two from 2021.

2) English language must be revised. For instance, at line 25 remove the Saxon genitive after the word “bridges”, at line 27 “has been used” should read “have been used”, and so on.

3) Conclusions look vague and too generic. Please add more details on the practical applications and the main final remarks (accuracy, precision, capabilities, etc.).

Authors are invited to submit a rebuttal version of the manuscript; the Reviewer’s decision is “Minor Revision”.

Author Response

The authors wish to thank the reviewer for the comments and for the timely and thoughtful review. Each comment is addressed individually below

Reviewer 1

This work proposes a literature review about the Digital Image Correlation in Structural Health Monitoring of bridge infrastructures.

General comments: the paper is well formatted; figures are clear (even if not all in hi-res); the topic fits well the purposes of the Journal.

The collection proposed by the Authors is of interest from an engineering standpoint and, as far as the knowledge of the Reviewer, the results are new and here published for the first time. Nevertheless, the following questions need to be carefully addressed by the Authors prior to publication.

Response: Agree. The authors wish to thank the reviewer for the comments and for the timely and thoughtful review.

1) The bibliography must be improved with recent works: out of 106 total articles, there are only four papers from 2020 and two from 2021.

- Response: New recent references have been added. The statistics of the references are illustrated in the following figure:

 No. of reference used in the paper is 112 

 2) English language must be revised. For instance, at line 25 remove the Saxon genitive after the word “bridges”, at line 27 “has been used” should read “have been used”, and so on.

- Response: Agree. The authors have revised it accordingly.

3) Conclusions look vague and too generic. Please add more details on the practical applications and the main final remarks (accuracy, precision, capabilities, etc.).

- Response: Agree. More DIC details and parameters were added and described based on the reviewed works, please refer to the review tracking markup for the conclusion section.

In addition, a summary of the reviewed works was presented in a separate table.

Reviewer 2 Report

In this review, the authors describe a modern, robust, and precise DIC method applied to estimate the damage degree of various bridge types. The authors compared the DIC method with conventional approaches and highlighted its limitations and advantages. This paper can be published after the following minor revision corrections:
1) All abbreviations should be denoted with capital letters like Digital Image Correlation (DIC) rather than Digital image correlation (DIC), e.g., line 22 (there are and other cases, please, check);
2) Please add scale bar for Figures 1, 2, 3;
3) Please separate Figure 4 on a) and b) parts. Moreover, try to improve its quality, e.g., improve resolution and make larger font;
4) Please do the same for Figures 5 and 6;
5) Please do not forget to define the abbreviation of 'LVDT';
6) Please improve the quality of Figures 7-13;
7) Please try to expand sections 3.1-3.4 with the following info: experiment and result description like the distance from the camera to object, camera characteristics, displacement/strain distributions, main observations, comparisons with other methods, errors, software for data post-processing, etc.;
8) line 513, correct typo 'The'.

Author Response

The authors wish to thank the reviewer for the comments and for the timely and thoughtful review. Each comment is addressed individually below

Reviewer 2

In this review, the authors describe a modern, robust, and precise DIC method applied to estimate the damage degree of various bridge types. The authors compared the DIC method with conventional approaches and highlighted its limitations and advantages. This paper can be published after the following minor revision corrections:

Response: Agree. The authors wish to thank the reviewer for the comments and for the timely and thoughtful review.

1) All abbreviations should be denoted with capital letters like Digital Image Correlation (DIC) rather than Digital image correlation (DIC), e.g., line 22 (there are and other cases, please, check);

Response: Agree. Authors checked and edited accordingly.

2) Please add scale bar for Figures 1, 2, 3;

Response: Agree. Added.

3) Please separate Figure 4 on a) and b) parts. Moreover, try to improve its quality, e.g., improve resolution and make larger font;

Response: Agree. Figure 4 is reconstructed with bigger details and font size.

4) Please do the same for Figures 5 and 6;

Response: Agree. Edited.

5) Please do not forget to define the abbreviation of 'LVDT';

Response: Agree. Defined accordingly, thank you for your notice.

6) Please improve the quality of Figures 7-13;

Response: Agree. The Figures 1-7 were completely replaced with new enhanced quality images produced or captured by the author himself. Figures 8-13 were eliminated due to quality and copyright issues.

7) Please try to expand sections 3.1-3.4 with the following info: experiment and result description like the distance from the camera to object, camera characteristics, displacement/strain distributions, main observations, comparisons with other methods, errors, software for data post-processing, etc.;

Response: Agree. All the mentioned parameters were demonstrated throughout the sections. However, sometimes the source work lack of these info, as a result the were not mentioned at some paragraphs.

8) line 513, correct typo 'The'.

Response: Agree. Corrected.

Reviewer 3 Report

Manuscript has been written in a very good manner, specially the selection of literature, which is very difficult in review papers, was made with sensitivity and knowledge. In my opinion, the work qualifies for publication in "Infrastructures". My only remark concerns the quality of the photos presented in the manuscript. Before publication, if possible, please provide photos with the better quality, especially this concerns to Figs. 1, 8, 9, 11 and 13.

Author Response

The authors wish to thank the reviewer for the comments and for the timely and thoughtful review. Each comment is addressed individually below

Reviewer 3

Manuscript has been written in a very good manner, specially the selection of literature, which is very difficult in review papers, was made with sensitivity and knowledge. In my opinion, the work qualifies for publication in "Infrastructures". My only remark concerns the quality of the photos presented in the manuscript. Before publication, if possible, please provide photos with the better quality, especially this concerns to Figs. 1, 8, 9, 11 and 13.

Response: Agree. The authors wish to thank the reviewer for the comments and for the timely and thoughtful review.

The photos are replaced accordingly. The Figures 1-7 were completely replaced with new enhanced quality images generated or captured by the author himself. Figures 8-13 were eliminated due to quality and copyright issues.